# Transfer learning efficiently maps bone marrow cell types from mouse to human using single-cell RNA sequencing

Patrick S. Stumpf [1,2✉], Xin Du[3], Haruka Imanishi[4], Yuya Kunisaki[5], Yuichiro Semba[6], Timothy Noble[1], Rosanna C. G. Smith [7], Matthew Rose-Zerili[7], Jonathan J. West [7,8], Richard O. C. Oreffo [1,8], Katayoun Farrahi [3], Mahesan Niranjan[3], Koichi Akashi[6], Fumio Arai [4✉] & Ben D. MacArthur[1,8,9,10✉]

Biomedical research often involves conducting experiments on model organisms in the anticipation that the biology learnt will transfer to humans. Previous comparative studies of mouse and human tissues were limited by the use of bulk-cell material. Here we show that transfer learning—the branch of machine learning that concerns passing information from one domain to another—can be used to efficiently map bone marrow biology between species, using data obtained from single-cell RNA sequencing. We first trained a multiclass logistic regression model to recognize different cell types in mouse bone marrow achieving equivalent performance to more complex artificial neural networks. Furthermore, it was able to identify individual human bone marrow cells with 83% overall accuracy. However, some human cell types were not easily identified, indicating important differences in biology. When re-training the mouse classifier using data from human, less than 10 human cells of a given type were needed to accurately learn its representation. In some cases, human cell identities could be inferred directly from the mouse classifier via zero-shot learning. These results show how simple machine learning models can be used to reconstruct complex biology from limited data, with broad implications for biomedical research.

[1] Centre for Human Development, Stem Cells and Regeneration, Faculty of Medicine, University of Southampton, Southampton SO17 1BJ, UK. [2] Joint Research Center for Computational Biomedicine, RWTH Aachen University, Aachen 52074, Germany. [3] Electronics and Computer Sciences, University of Southampton, Southampton SO17 1BJ, UK. [4] Kyushu University, Department of Stem Cell Biology and Medicine, Graduate School of Medical Sciences, Kyushu University, Fukuoka 812-8582, Japan. [5] Center for Cellular and Molecular Medicine, Kyushu University Hospital, Fukuoka 812-8582, Japan. [6] Department of Medicine and Biosystemic Science, Kyushu University Graduate School of Medical Sciences, Fukuoka 812-8582, Japan. [7] Cancer Sciences, Faculty of Medicine, University of Southampton, Southampton SO16 6YD, UK. [8] Institute for Life Sciences, University of Southampton, Southampton SO17 1BJ, UK. [9] Mathematical Sciences, University of Southampton, Southampton SO17 1BJ, UK. [10] The Alan Turing Institute, London NW1 2DB, UK. ✉email: pstumpf@ukaachen.de; farai@med.kyushu-u.ac.jp; bdm@soton.ac.uk

The translational biomedical research pipeline typically consists of a sequence of phases that starts with a discovery phase, which usually involves experiments on cell lines cultured in vitro as well as in vivo studies in model organisms, and ends with carefully controlled clinical, review, and monitoring phases[1]. The eventual success of this pipeline depends upon effective transfer of information from one phase of the process to the next. Despite the tremendous cost associated with translational research failure[2], this information-transfer process is poorly understood.

Transfer learning is the branch of machine learning that takes information derived from one setting and applies it to improve generalization in another area[3]. The basic idea of transfer learning is to mimic the human ability to learn new concepts from limited examples by associating new information with prior understanding. In the transfer learning process, information gained from solving a problem in a source domain is passed to another related problem in a target domain, thereby improving target domain performance. The gain from such knowledge transfer is particularly apparent whenever data are abundant in the source domain but scarce in the target domain. In this case, new concepts can be effectively learnt in the target domain from very few training samples, via leveraging of prior knowledge.

Here, we show how transfer learning can be used to map bone marrow biology from mouse to humans. A number of previous studies have conducted inter-species comparison of gene expression profiles at the tissue level[4–7]. The novelty of our approach includes the use of single-cell data, thereby achieving a substantial improvement in resolution over previous studies. The problem of passing information from a model organism (the source domain, here the mouse) to the humans (the target domain) was chosen because it is central to successful translational research. Bone marrow was chosen because it is a complex tissue, consisting of numerous different cell types, present in differing proportions, with a well-established physiology in mouse that is broadly conserved, and yet only partially understood, in humans. We demonstrate how using a machine learning model to encode cell-type information in mouse enables cell-type comparison with humans, providing new insight into the effective transfer of information between species and the amount of domain-specific information that is required to train a machine learning model using single-cell data.

## Results

**Mapping mouse bone marrow.** To begin, we collected gene expression signatures using droplet-based scRNA-Seq (Drop-Seq[8]) from unfractionated total bone marrow (TBM) samples as well as from weakly lineage-depleted bone marrow (DBM) Cd45/Ter119 dual-negative subsets in order to enrich for rarer cell types, from three different mice (Fig. 1a). Overall, 6800 single-cell transcriptomes were sequenced, yielding greater than $9 \times 10^4$ reads per cell on average. Following pre-processing and filtering, a total of 5504 cells were retained, expressing on average 2684 transcripts per cell.

We then performed unsupervised clustering using the Louvain[9] method (see "Methods") to identify the various hematopoietic and niche-cell types present. Despite the sparsity ($93.3 \pm 0.6\%$; mean ± s.d. in mouse; Supplementary Fig. 1a) and substantial technical variability that is typically encountered in scRNA-seq data[10], we found that cells clustered according to their type, rather than the mouse from which they were obtained (Fig. 1b), suggesting the presence of a common and robust "map" of the mouse bone marrow (Fig. 1b–d and Supplementary Figs. 1b, d, and 2)

Assignment of cell identities to clusters was performed by examining the localization of established lineage markers to distinct clusters (see Fig. 1f, Supplementary Fig. 2 and

"Methods"). Our cluster annotation was in accordance with other recent publications[11,12]. In total, we identified 19 cell populations, covering the erythroid, myeloid, and lymphoid branches of hematopoietic lineage tree, as well as separate populations of non-hematopoietic supporting cell types including endothelial cells and pericytes (Fig. 1c–f).

Four features of this clustering are notable. Firstly, the proportion of cells in each cluster varied considerably, reflecting the balance of different cell types present in the mouse bone marrow (Fig. 1e). Clusters associated with rare cell types, such as hematopoietic stem and progenitor cells (HSPCs), contained very few cells. In contrast, clusters associated with abundant cell types, such as erythroid cells, contained large numbers of cells. To gain resolution on rare/immature cell types the depletion protocol we used reduced the relative abundance of various mature cell types —including monocytes ($-8.1 \pm 3.2\%$ relative to TBM; mean ± s.d. from $n = 3$ biological replicates), myelocytes ($-11.4 \pm 5.6\%$ relative to TBM), pro-B lymphocytes ($-9.4 \pm 3.8\%$ relative to TBM), while neutrophils, pre-B-lymphocytes, and T-lymphocytes were completely ablated—and enriched for immature cells including HSPCs ($+4.0 \pm 0.9\%$ relative to TBM), myeloblasts ($+7.5 \pm 3.7\%$ relative to TBM), monoblasts ($+2.8 \pm 1.7\%$ relative to TBM), erythroblasts ($+5.4 \pm 1.9\%$ relative to TBM), pericytes ($+0.9 \pm 0.3\%$ relative to TBM), and endothelial cells ($+2.1 \pm 0.5\%$ relative to TBM).

Secondly, while some clusters represent distinct cell identities, others are associated with the discretization of continuous maturation processes. For example, the erythroblast cluster (shown in red in Fig. 1c) consists of a heterogeneous mixture of cells at different stages in the erythrocyte maturation process, representing a gradual transition from immature pro-normoblast to late normoblast (see Supplementary Fig. 2a).

Thirdly, it is well-established that cell types in the hematopoietic cell lineages of the bone marrow are arranged according to a hierarchical structure[13–15]. By considering adjacency relationships between clusters, we were able to broadly recapitulate the known structure of this hierarchy, indicating that the clustering structure that we observed captures salient features of the mouse bone marrow biology (Fig. 1d).

Fourthly, identified clusters are visually separable in two-dimensional embeddings of the data generated using nonlinear methods such as t-distributed stochastic neighbor embedding (tSNE, Fig. 1c) and uniform manifold approximation and projection (Supplementary Fig. 1b), yet they are not easily separable in using linear methods such as principal component analysis (Supplementary Fig. 1d), suggesting that genomic features combine in a nonlinear way to define cell identities.

Collectively these clusters, and the spatial relationships between them, constitute a reference map of the mouse bone marrow. However, this map is not in a form directly amenable to comparison with human bone marrow. To allow comparison we trained a multinomial logistic regression (MLR) model to classify individual cells from their gene expression profiles (using the unscaled, binarized data, see "Methods"). MLR, a multiclass generalization of logistic regression, was chosen since it is a simple generalized linear method (i.e. it makes predictions based on linear combinations of inputs via a nonlinear output function) that has been shown to be as powerful as more complex machine learning methods in other biomedical contexts[16] while maintaining superior interpretability. Since we ultimately wanted to compare this map with a similar map of human bone marrow, we restricted our analysis to those genes with a human orthologue (specifically we included genes that had a unique human homolog and exhibited high expression variability in mouse or human bone marrow samples, see "Methods"). Because cell identities were determined from unsupervised clustering of the data, this is

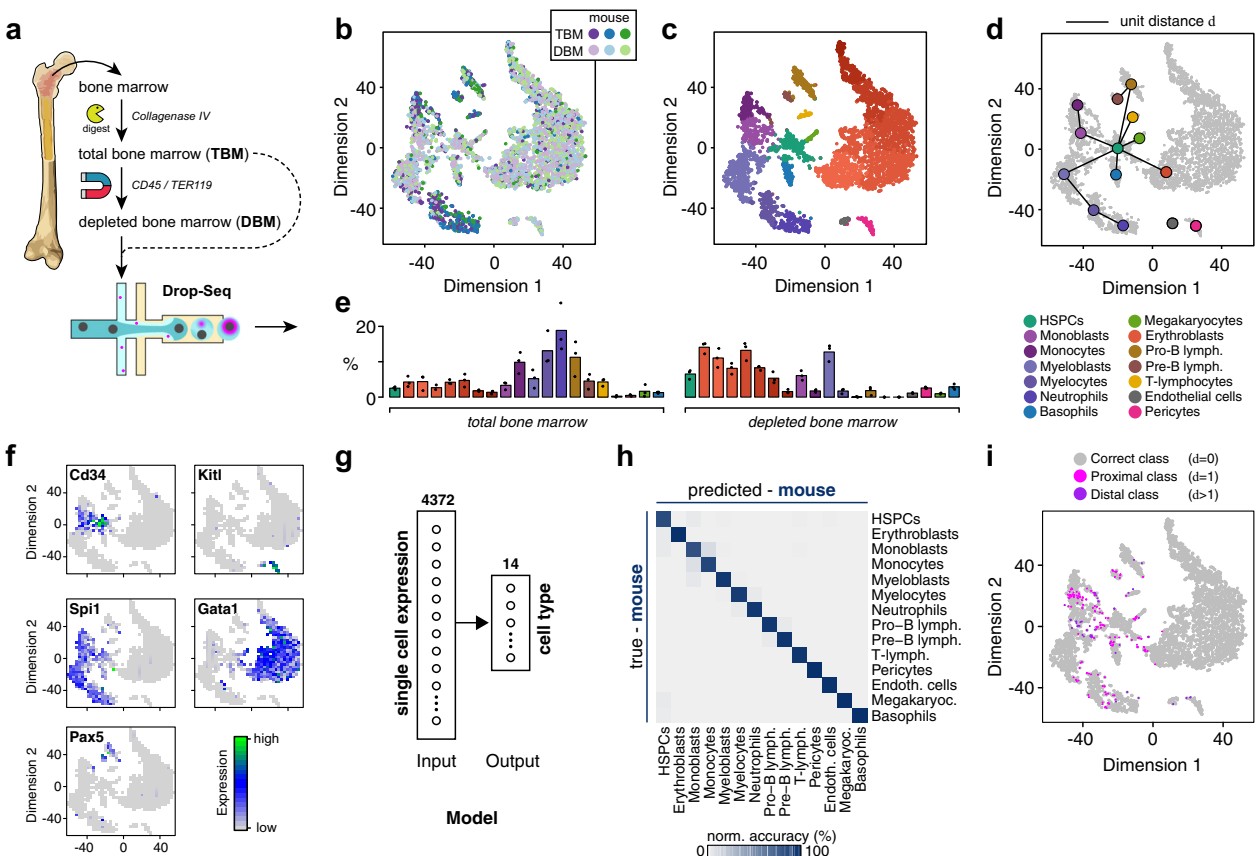

**Fig. 1 Dissecting the cellular heterogeneity of the mouse bone marrow. a** Experiment schematic. Single-cell RNA sequencing was performed on total and depleted (CD45−/Ter119−) bone marrow cells. **b** Projection of data onto two dimensions using *t*-distributed stochastic neighbor embedding (tSNE[63]) indicates that bone marrow population structure is preserved in biological replicates (*n* = 3, shown in purple, blue, and green). **c** Cell types were identified using unsupervised clustering[9] followed by annotation of clusters according to localization of known markers for different cell types. **d** Clusters naturally arrange in accordance with the known bone marrow lineage tree. The lineage tree shown is taken from ref. [13]. HSPCs hematopoietic stem and progenitor cells. **e** Relative abundance of cell types in total and depleted bone marrow samples. Bar height indicates the mean over the biological replicates (*n* = 3). **f** Key markers of the main branches of the hematopoietic lineage tree and niche cells localize to distinct clusters in the data. The following representative markers are shown: stem and progenitor cells: Cd34; niche cells: Kitl; myeloid lineage: Spi1; erythroid lineage: Gata1; lymphoid lineage: Pax5. See Supplementary Fig. 2 for localization patterns of a range of other markers. **g** Schematic of the multinomial logistic regression (MLR) model used to identify cell types from gene expression profiles obtained from mouse bone marrow cell samples. The MLR consists of an input layer with 4372 units (corresponding to the set of high variability genes that have unique human homologs; see Methods), and a 14-class SoftMax output layer. **h** Confusion matrix of validation data, showing accurate classification of cell identities by the ANN. Data displayed are the average over a fivefold cross-validation. **i** Distribution of misclassified cells in the training data. Color represents the distance *d* between the true and predicted label in the cell lineage tree in panel **d**.

an easy learning problem readily solved by the MLR (Fig. 1g–i). The resulting model performed well, achieving average balanced classification accuracy of 97.0 ± 0.4% (mean ± standard deviation over fivefold cross-validation), and was able to reliably identify cells of every type (Fig. 1h).

Notably, misclassification was largely constrained to cells in proximity to cluster boundaries imposed along continuous developmental trajectories (Fig. 1i). To dissect this observation further we systematically investigated misclassification by taking advantage of the fact that the classes in the mouse training data are arranged according to a biologically meaningful hierarchy that encodes the bone marrow lineage tree (see Fig. 1d). Whenever misclassification of a cell occurred, we determined the relationship between its true class and its (falsely) predicted class. We denoted a misclassification to be proximal if the predicted class is immediately adjacent to the true class in the lineage tree and distal otherwise. Overall, a low incidence of proximal misclassification was observed (3.1 ± 0.26%; mean ± s.d., *n* = 5), while distal misclassification occurred even more rarely (0.9 ± 0.35%; mean ± s.d., *n* = 5).

Moreover, patterns of misclassification were not uniform. For example, cells in the HSPC cluster were most likely to be misclassified (proximal: 8.9 ± 2.5%; distal: 0.7 ± 0.9%; mean ± s.d., *n* = 5). This is likely partly due to the limited number of HSPC training samples available. It also occurs because HSPCs are a heterogeneous population with expression patterns that partially intersect with several other cell types. This observation agrees with recent studies in mouse, humans, and zebra fish, which have shown that the HSPC pool is a particularly variable cell population[12–14,17,18], and highlights the fact that classification accuracy will depend upon both the amount of data available for training and the heterogeneity intrinsic to the cell population being considered.

The MLR classifier assigns cellular identities by passing linear combinations of gene expression patterns through a softmax output function. The input weights to the softmax function therefore constitute a simple way to identify subsets of genes that are most strongly associated with each cell identity (see "Methods"). This analysis recapitulated well-established

molecular markers of bone marrow cell-type identities (see Supplementary Data 1).

For instance, among the top-ranking features associated with the HSPC identity are Angpt1 and Myct1, both known regulators of stem cell proliferation[19–21]; Irf8, a monocyte lineage determinant[22], was most strongly associated with the monoblast identity; Ccr2 and Ctss, which are known to play a central role in chemotaxis and antigen presentation of monocytes[23,24], were influential for monocyte classification. Assignment of the myeloblast identity was highly sensitive to Prtn3 (also known as Myeloblastin), while transcripts encoding components of secretory vesicles that are sequentially produced during myelopoiesis[25], and define the morphologically distinct stages of myeloblasts (primary granules; Elane), myelocytes (secondary granules; Ltf), and neutrophils (tertiary granules containing the neutrophil-collagenase Mmp8) also strongly influenced these class assignments. Similarly, various different immature lymphocytes were discriminated based on the expression of Vpreb3 (pro-B-cells), Cd74 (pre-B-cells), and Cd3d (T-lymphocytes), while (peri-)vascular cells were determined based on Gpx8 (pericytes) and Kdr (endothelial cells) expression among other genes. Supplementary Data 1 contains a list of all the top-ranking genes associated by the MLR with each cell type.

Collectively, these results indicate that a simple machine learning model is able to reconstruct cellular identities from patterns of gene expression based upon easily interpretable and biologically plausible mechanisms. In order to understand whether classifier performance could be improved we also compared performance of the MLR model to a feedforward artificial neural network (ANN) classifier (Supplementary Fig. 3a). We chose a feedforward ANN because it can be considered as a generalization of an MLR in which the input signal is passed through hidden layers, which potentially allows it to identify more complex relationships[26] involving nonlinear interactions between genes. We found that optimal ANN performance was achieved by introducing a single hidden layer of 16 neurons. Overall, the resulting ANN has 70,206 parameters, compared to 61,222 for the MLR and so does not constitute a substantial increase in model complexity.

As expected, the ANN (Supplementary Fig. 3a–c) performed as well as the MLR, achieving balanced accuracy (BA) of $96.7 \pm 0.9\%$ (mean ± s.d. from fivefold cross-validation). However, while the MLR decision-making process can be easily reconstructed based upon a single set of weights, that of the ANN cannot. To dissect the biological basis of the ANN performance we therefore conducted a sensitivity analysis designed to determine subsets of genes that are most strongly associated by the ANN with each cell identity (see "Methods" for details and Supplementary Data 2 for a list of top-ranking ANN features). We found a strong overlap in the sets of important genes identified by the ANN and MLR (see Supplementary Fig. 3d), indicating that the two models classify cells on the basis of similar biological criteria.

To investigate the functional significance of these gene sets we also performed Gene Ontology (GO) term analysis (see "Methods"). We found that significantly enriched GO terms associated with these gene lists summarized the biological function of their associated cell type. For the ANN key GO associations included *hemopoiesis* for HSPCs ($p = 8.5e-5$; modified Fisher's exact test), *blood coagulation* for megakaryocytes ($p = 1e-8$), *B cell receptor signaling* for pro-B- and pre-B-cells ($p = 1.2e-8$; $p = 3.9e-10$); *T-cell receptor signaling* for T-lymphocytes ($p = 1.6e-10$); *cell adhesion* and *osteoblast differentiation* for pericytes ($p = 7.6e-10$; $p = 9.5e-9$); *cellular response to VEGF* for endothelial cells ($p = 8.4e-8$); *positive regulation of mast cell degranulation* for basophils ($p = 1.2e-6$); and *innate immune response* and related terms for monocyte- and granulocyte lineages. Supplementary Data 3 contains a complete list of GO terms associated with each cell type by the ANN (see also Supplementary Data 4 for similar GO term analysis of MLR weights).

Collectively, these results indicate that both the MLR and ANN models capture the essential biology of the mouse bone marrow and can accurately discriminate between mouse bone marrow cell types based upon differences in biologically significant gene expression patterns.

**Mapping human bone marrow**. We next sought to determine the extent to which the biology learnt in the mouse "source" domain could be transferred to the human "target" domain of true interest. To do this, we sequenced bone marrow samples from three patients undergoing routine hip replacement surgery at Southampton General Hospital. In total, ~25,000 single-cell transcriptomes from three patients were sequenced yielding on average $5 \times 10^4$ reads per cell. As with the mouse, we sequenced unfractionated bone marrow as well as depleted populations in order to enrich for rarer cell types. Following pre-processing and filtering of low-quality cells (see "Methods") we obtained data for 9394 cells expressing on average 3070 transcripts per cell, corresponding to a data sparsity of $95.5 \pm 0.95\%$ mean ± s.d. (Supplementary Fig. 1a). As with the mouse data we then performed unsupervised clustering to identify the various hematopoietic and niche-cell types present and assigned cell identities based upon localization of established lineage markers (see Supplementary Fig. 4 and "Methods: Human bone marrow cell characterization"). As with the mouse data this analysis resulted in a set of single-cell transcriptomes in which each cell is annotated with a unique identity determined by unsupervised clustering.

We subsequently assessed the extent to which our mouse MLR and ANN classifiers, which were trained exclusively on mouse data, were able to predict human cell identities (Fig. 2a). We found that the mouse-trained MLR predicted human cell identities remarkably well, achieving an average BA of 82.7%. The ANN model performed negligibly better at 83.3% average BA, see Supplementary Fig. 3f. Notably, this overall accuracy was not consistent across all cell classes: rather, accuracy ranged from 63.0 to 99.9% for individual cell classes (Fig. 2b). The ANN model had a very similar range of accuracy (60.0–98.0%, Supplementary Fig. 3f). Thus, while some human cell types were identified remarkably well by the mouse classifier, indicating strongly shared biology, others cell types were much more poorly aligned, indicating systematic differences in underlying biology between the species. For example, human erythroblasts and T-lymphocytes were rarely misclassified by the mouse model (which achieved 97.4% and 99.3% BA in identifying these classes, respectively; the corresponding values of the ANN are 97.5% and 98.0%), while other cell types were frequently misclassified (Fig. 2b).

As with the mouse data, we found that misclassification of human cells was commonly proximal in nature (15.7% versus 12.0% for distal misclassification; the corresponding values of the ANN are 13.9% proximal and 8% distal), suggesting that the mouse classifiers had partially learnt human cellular identities, and misclassification was not entirely artefactual (Fig. 2b–d). For instance, human HSPCs were systematically misclassified as one of their proximal descendent classes (18.4% proximal misclassification; 14.0% in case of the ANN), but less frequently distally misclassified (2.5% distal misclassification; 5.4% in case of the ANN) (Fig. 2b–d and Supplementary Fig. 3f–h). A similar pattern of misclassification of mouse HSPCs was also seen (Fig. 1i). Likewise, myelocytes were systematically misclassified as their progenitors, myeloblasts, or their descendents, neutrophils (MLR:

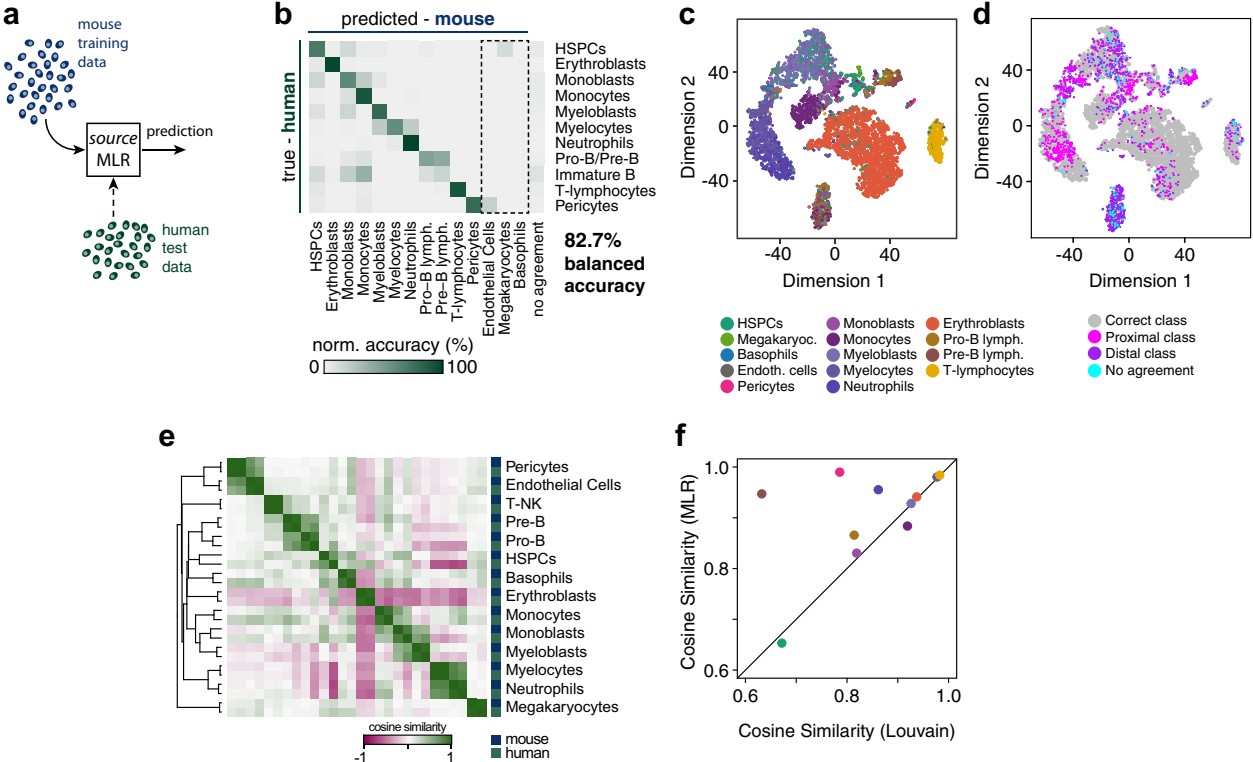

**Fig. 2 Bone marrow biology maps partially from mouse to humans. a** Schematic of the naïve transfer process. The MLR trained in the source domain (mouse) is used to classify test data from the target domain (humans). **b** Confusion matrix of classification consensus from fivefold cross-validation. The dashed box highlights cell types identified in the mouse but not the human data. **c**, **d** Projection of human data onto two dimensions using tSNE[63]. Points represent cells colored by **c** predicted cell identity or **d** misclassification. Cells for which the five classifiers did not agree are shown in turquoise. **e** Heatmap of similarity between mouse and MLR predicted human cell types. Similarities were calculated between cell-type mediancentres[27] using cosine similarity (see "Methods"). Clustering using single linkage reveals high similarity between equivalent mouse and human cell types. **f** Similarity of mouse and human cell types using annotations obtained from unsupervised Louvain clustering (x-axis) and MLR model predictions (y-axis). Pericytes (pink) and Pre-B-lymphocytes (brown) contain a mixture of cell types that are not resolved by unsupervised clustering but are identified by the MLR model. The black diagonal marks $y = x$.

39.8% proximal versus 4.7% distal misclassification; ANN: 29.6% proximal versus 8.7% distal misclassification) (Fig. 2b–d and Supplementary Fig. 3f–h). However, this pattern was not universal.

For example, while human immature B lymphocytes were commonly misclassified as their progenitors, pro-B and pre-B lymphocytes, they were also systematically misclassified as a range of other types of progenitors (MLR: 10.4% proximal versus 67% distal misclassification; ANN: 14.1% proximal misclassification versus 33.8% distal misclassification; see Fig. 2b–d and Supplementary Fig. 3f–h), indicating that neither of the mouse classifiers was able to fully resolve the human immature B cell identity. Notably, the mouse data did not contain a comparable immature B cell cluster, and so the mouse model was never explicitly trained to recognize the expression signatures of B lymphocytes. Nevertheless, the mouse ANN, but not the MLR, assigned the majority of immature B lymphocytes to adjacent clusters and hence to the correct branch of B cell development.

Taken together, these results indicate that while much biology is conserved between the mouse and human bone marrow, there are systematic differences in expression patterns. These differences are important because they indicate the circumstances in which the mouse is likely to be a good model of human biology and when it will likely not, and they highlight instances where a comparison is not immediately possible.

To investigate these differences further, we assessed the similarity of mouse and human cell types directly from their

transcriptional profiles. To do so, we first established a single characteristic gene expression pattern for each cell type by calculating the mediancentre[27] (a multivariate generalization of the median) of all cells associated with that type. Mouse cell annotations were obtained from unsupervised clustering of the mouse data (as described above, see Fig. 1c and Supplementary Fig. 2). Human cell annotations were either (1) predicted from the mouse MLR/ANN model or (2) obtained from unsupervised clustering of the human data (as described above, see Supplementary Fig. 4). We then calculated distances between pairs of mediancentres (see "Methods" for further details) to determine similarities of cell types. In accordance with our machine learning results, this analysis revealed a high degree of similarity between equivalent mouse and human cell types (Fig. 2e, f). Furthermore, the annotation provided by the machine learning models was able to resolve some populations that were not apparent using unsupervised clustering. For instance, labeling of human Pre-B cells and Pericytes using the MLR identified populations of cells that were substantially more similar to their mouse counterparts than labeling by unsupervised clustering (Fig. 2f), indicating that the MLR was able to resolve more defined populations of cells than naïve clustering alone.

**Discovery of hidden cell identities using zero-shot learning.** While the mouse and human datasets contain data from many of the same cell types, some cell types were not resolvable in the human samples with accuracy comparable to the mouse. In

particular, we could not identify distinguishable clusters associated with endothelial cells or megakaryocytes, yet both of these cell populations were clearly apparent in the mouse data. Because many aspects of bone marrow are conserved between mouse and humans, we next sought to determine if the mouse model could be used to help resolve the biology of such hard-to-identify human cell types.

Notably, a substantial subset of human HSPCs were identified as megakaryocytes by the mouse classifiers (MLR: 10.8%; ANN: 19.4%, see Fig. 2b, c and Supplementary Fig. 3f, g). This high overlap is notable because HSPCs and megakaryocytes are proximal in the mouse hematopoietic cell lineage map (Fig. 1d), reflecting the fact that megakaryocytes emerge directly via differentiation from HSPCs[13,28,29]. This result suggested to us that the mouse classifiers might be revealing aspects of human HSPC/megakaryocyte biology that are not apparent from unsupervised clustering of the human dataset alone. To investigate these differences further, we conducted sensitivity analysis (see "Methods") to identify the genes that carry the most discriminatory information in distinguishing between megakaryocytes and HSPCs in the mouse ANN classifier (the ANN was chosen for this analysis since it classified a larger percentage of HSPCs as megakaryocytes than the MLR; a similar pattern was also seen on classification of pericytes/endothelial cells, see below). Examination of co-expression patterns of these genes in human and mouse cells confirmed that megakaryocytes are characterized by broadly similar expression signatures in both mouse and humans and are distinguishable from HPSCs based on these expression patterns (Supplementary Fig. 3i).

Notably, both mouse and human megakaryocytes expressed high levels of transcripts involved in platelet biogenesis such as Rab27b[30], Ppbp[31], and in platelet function (hemostasis) such as Itga2 (ref. [32]) (encoding collagen receptor CD49b) and F2rl2 (ref. [33]) (encoding coagulation factor 2) (Supplementary Fig. 3i). Similarly, HSPCs in both species shared expression of key transcription factors such as Zfp36l2 and Sox4 (Supplementary Fig. 3i) that are known to control stem cell self-renewal[19,34].

Likewise, a substantial subset of human pericytes were identified as endothelial cells by the mouse classifiers (MLR: 17.2%; ANN: 20.3%, see Fig. 2b, c and Supplementary Fig. 3f, g). When we examined the clustering structure of the mixed population of mouse and human pericytes/endothelial cells, we observed that a subset of human pericytes clustered with mouse endothelial cells (Supplementary Fig. 3j). While the ontogeny of pericytes and endothelial cells in the adult bone marrow is currently unclear[35], both cell types are constituents of the vasculature, and are in close spatial proximity in the bone marrow. Taken together these results suggest that the mouse classifier might be revealing aspects of human biology that are not apparent from unsupervised analysis of the human dataset alone.

To investigate these differences further we again conducted sensitivity analysis (see "Methods") to identify the genes that carry the most discriminatory information in distinguishing between endothelial cells and pericytes in the mouse ANN model. Among the genes that were identified were a number of important endocrine modulators and sensors of energy homeostasis such as Igfbp5 and Lepr[36,37]; paracrine signaling molecules such as Cxcl12 (ref. [38]); and components of the iron cycle such as Cp. Examination of co-expression patterns of these genes revealed a substantial overlap between mouse and human pericyte expression patterns, indicating that much of the central molecular machinery of these cells is evolutionarily conserved (Supplementary Fig. 3j). Similarly, both human and mouse endothelial cells shared expression of known angiogenic-signal receptors such as Kdr[39] and the Vegf target gene Fabp4[40] (Supplementary Fig. 3j), again highlighting shared biology.

Collectively these results indicate that once encoded in a machine learning model, mouse data can be used to contextualize human data, identify evolutionarily conserved gene expression patterns, and thereby provide insight into poorly resolved cell populations. In the machine learning literature, the process of object identification without training examples is known as zero-shot learning, and typically relies on importing prior knowledge from a related source domain[41]. Here, because the mouse classifier encodes evolutionary conserved information, it can be used, in conjunction with prior knowledge of the bone marrow lineage tree, to infer poorly resolved human cell populations via a process of biologically-guided zero-shot learning.

**Transferring biology from mouse to humans.** Since the mouse classifiers were not able to accurately identify all human cell identities, yet appeared to be capturing aspects of evolutionarily conserved biology we next sought to determine if they could be used to train a more accurate model of the human bone marrow. To achieve this objective, we re-trained the mouse classifiers using a limited set of human bone marrow cell gene expression signatures as additional training data (Fig. 3a).

We produced a series of revised classifiers by re-training the mouse classifiers using increasing numbers of additional human training examples (see Fig. 3a for a schematic of the MLR re-training and Supplementary Fig. 5a for a schematic of the ANN re-training). In both cases, classification performance sharply increased on re-training, even when only a very small number of representative human training examples were used (Fig. 3c and Supplementary Fig. 5c). Notably, classifier performance began to saturate when re-training using 4–8 additional human training examples for each class (44–88 cells in total). At this point re-trained models achieved over 90% BA (up from 82.7% in the naïve mouse MLR; 83.3% in the naïve mouse ANN) and a significantly improved F1 score (MLR: 81.6 ± 0.6% up from 64.0 ± 1.0%; ANN: 83.2 ± 0.8% up from 63.1 ± 0.6%; mean ± s.d., $n = 5$), indicating that human cell identities can be reliably encoded upon re-training of the mouse models with very few training examples (Fig. 3c and Supplementary Fig. 5c).

In order to assess the extent to which pre-learning in the mouse source domain improved classification performance in the human target domain, an equivalent set of models were trained without transfer from the mouse (i.e. from randomized initial conditions, referred to as naïve models; Fig. 3a–c and Supplementary Fig. 5a–c, g, h). Since they benchmark the efficiency with which human bone marrow biology can be learnt from low volumes of data without pre-training in the mouse, these naïve models act as controls for the transfer learning process.

To determine the efficiency of the information-transfer process we plotted classifier performance (here, the F1 score, which accounts for both the precision and recall of the classifiers) of transferred and naïve models against each other as the number of human training examples varied, to produce characteristic learning curves (Fig. 3d, e for the MLR and Supplementary Fig. 5d, e for the ANN). This analysis quantifies the extent to which the biology of each cell type is shared between species (see Fig. 3d and Supplementary Fig. 5d and further explanation in Fig. 3e). Four distinct groups of cell types can be distinguished based on their different characteristic learning curves (Fig. 3d and Supplementary Fig. 5d).

The first group contains cell types with highly conserved phenotypes, which display high classification performance initially (i.e. the mouse classifier is able to identify human cells without additional training using human data) that does not improve considerably upon re-training in the target domain (i.e. using additional human data; Fig. 3d–f and Supplementary

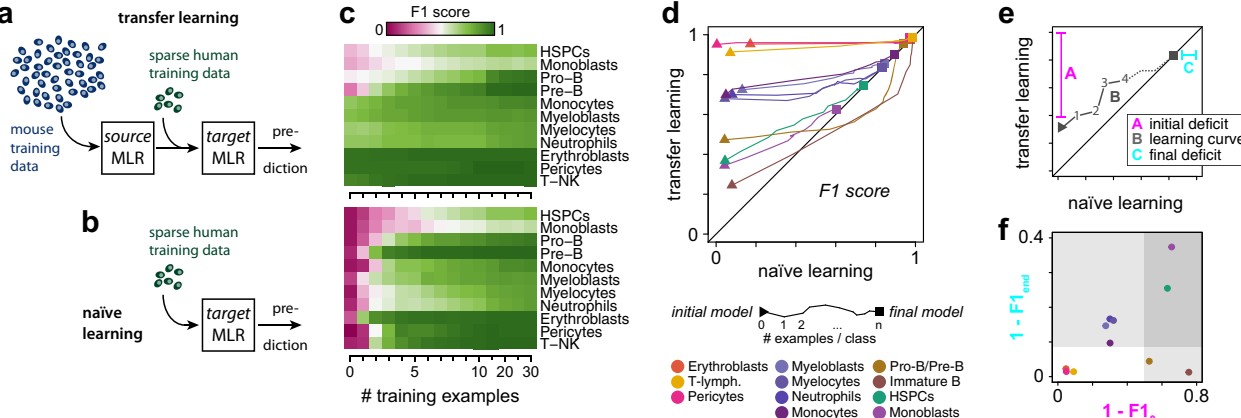

**Fig. 3 Mapping biology from mouse to human using transfer learning. a** Schematic of the transfer learning process. Abundant data from source domain (here the mouse) are used to train a source MLR. Sparse data from the target domain (here the humans) is used to fine-tune the parameters of the source MLR, thereby transferring knowledge from source to target domain. **b** Schematic of naïve learning as a control for transfer learning. Rather than updating the pre-trained mouse model, a series of separate MLRs are trained from random initial conditions on sparse data from the human target domain. **c** Both transfer and naïve learning improves with the number of human samples used for training (shown is data for 0,1, 2, …, 10, 15, 20, 25, 30 human cells per class). Transfer learning performance (top) and naïve learning (bottom). Displayed is the average $F1$ score from fivefold cross-validation as a measure of classifier performance. **d** Learning curves illustrate the evolution of classification performance starting from the initial mouse (triangle) to the final model (square; trained on 30 human examples per class). **e** Schematic to interpret the learning curves in panel **d**. Three features are of importance. $A$ is the initial performance deficit = $1 - F1_0$, where $F1_0$ is the $F1$ score of the mouse model in predicting human samples. $B$ is the learning curve: each point on this curve plots the $F1$ score of the re-trained mouse model against the naïve human model for a fixed number of human training examples from 0 to 30 per class. $C$ is the final performance deficit = $1 - F1_{end}$, where $F1_{end}$ is the $F1$ score of the naïve human model trained on 30 samples per class in predicting human samples. The line $y = x$ is in black. On this line the naïve and re-trained models have equivalent accuracy for the same number of human training samples. At this point the advantage of transfer learning is neutralized, and equivalent learning can be achieved by a naïve human model. All learning trajectories eventually converge to this line. **f** Cell types may be grouped by their initial and final performance deficits. Equivalent re-training results for the ANN classifier are shown in Supplementary Fig. 5.

Fig. 5d–f). This group includes erythroblasts and T-lymphocytes. These cell types: (1) are highly homogeneous in their expression patterns in humans and thus are consistently classified; and (2) have a biology that is highly conserved between mouse and humans. These cell types can be reliably identified from the mouse classifier and do not require any human training data to learn their representation. The mouse is a good model of human biology for these cell types with respect to their gene expression signatures.

The second group contains cell types that display good (but not excellent) classification performance initially, which does not improve considerably upon re-training with human data (Fig. 3d–f and Supplementary Fig. 5d–f). The second group includes pericytes, myeloblasts, myelocytes, neutrophils and monocytes. These cell types: (1) are more heterogeneous in their expression patterns in human and thus are less consistently classified than group 1 and (2) have a biology that is highly conserved between mouse and humans. This group of cell types requires a moderate amount of human training data to learn their representations.

The third group contains cell types that display initially low classification performance that improves rapidly upon re-training with human data (Fig. 3d–f and Supplementary Fig. 5d–f). This group contains pro-B/pre-B lymphocytes and immature B lymphocytes. These cell types: (1) are homogeneous in their expression patterns in humans; yet (2) have a biology that is distinct between species. The biological differences between species for these cell types are likely in part due to differences in cluster definition in mouse and human data. Specifically, while the mouse classifiers were trained to distinguish pro-B and pre-B cells, these cells are part of the same cluster in the human data. Hence, re-training involves separating the joint cluster of pro-B and pre-B lymphocytes from previously unseen immature B

lymphocytes. This group of cell types requires a moderate amount of human training data to learn their representations.

Finally, the fourth group contains cell types that display low classification performance initially that does not improve considerably upon re-training with human data (Fig. 3d–f and Supplementary Fig. 5d–f). This group contains monoblasts and HSPCs. These cell types: (1) are heterogeneous in their expression patterns in humans and (2) have a biology that is distinct between species. This group of cell types requires a larger amount of human training data to learn their representations.

Collectively, this analysis shows how tools from transfer learning can be used to dissect those aspects of biology that effectively transfer between the species and those aspects that do not.

## Discussion

Successful biomedical research is dependent on the effective transfer of information between different stages of the research pipeline. A critical step in this process is the transfer of biology from model organisms to the humans. Here, we have shown how methods from transfer learning can be used to pass biological information between species, using the bone marrow as an example. As increasingly detailed single-cell maps of whole organism biology become more available, we anticipate that transfer learning approaches will provide essential tools for comparative physiology.

We find that the mapping from gene expression patterns to cell types can be learnt using a simple generalized linear model (here, MLR) that is easily interpretable. This learning is only marginally improved by more complex artificial neural networks, which suffer from poor interpretability. Our results therefore highlight the importance of considering both performance and interpretation when constructing machine learning models and warn

against the use of unnecessarily complex methods. However, while source domain (i.e. mouse) performance was comparable for MLR and ANN models, we did find that ANN learning transferred slightly better than the MLR to the target domain (i.e. humans), as indicated by generally lower overall misclassification rates and higher proximal misclassification rates. This is likely because the additional hidden layer in the ANN facilitates generalization, as has been observed in other contexts[42]. In this instance the modest benefit is not justified. We anticipate that investigation of learning strategies that can improve generalization by systematically encoding shared and species-independent biological features in different aspects of network architecture—for example, by leveraging prior knowledge, such as known differences in molecular regulatory network architectures—may be a fruitful avenue for future research, although care should be taken to avoid overfitting. In this instance the marginal gains afforded by the ANN do not outweigh the increased model complexity or interpretability.

It is important to note that our approach has some limitations. Firstly, similarity of gene expression patterns only partially explain similarity in biology: epigenetic factors, not directly encoded in transcriptional patterns, are also likely to have an important role. Furthermore, here we have used orthologous genes, i.e., the genes that are shared between species (and have not multiplied) to compare biology. However, numerous aspects of biology are regulated by non-orthologous genes and so are not considered in our results. Moreover, the discretization process we have employed may not capture important differences in expression dynamics. During speciation, homologs may evolve in subtle ways, such that the resulting protein displays species-specific affinity or reaction rates or binding partners, which determine subtle differences in gene function. Such considerations have not been addressed here, although they could also be approached using similar transfer learning methods.

Previous studies have investigated the conservation of transcriptome signatures in tissues from multiple species using bulk mRNA sequencing[4–7]. The main limitation of these data is that mRNA derived from tissues consists of a mixture of cell types of unknown proportions. The single-cell approach described here overcomes this limitation, enabling analysis of tissue similarity at the level of cell types via a combination of unsupervised and supervised machine learning. Moreover, previous studies have relied on principal component analysis (PCA) to compare the localization of tissue samples in the first principal components, excluding higher principal components from analysis. However, since substantial variability typically is contained in more than two principal components, this led to conflicting results, with some studies reporting dominant species-specific differences[7], and other studies dominant tissue-specific differences[4–6]. Indeed, a meta-analysis of data from these data, using consistent pre-processing and a range of pairwise sample distance metrics instead of PCA, concluded that differences between tissues are greater than between species[43]. Here, using single-cell data, we observe that the largest source of variation in combined mouse and human data is related to cell type and not species (Fig. 2e). While this study focusses on a comparison of mouse and human bone marrow biology, recently, other studies have conducted comparisons of pancreas[44] and lung cancer[45]. These studies also demonstrated that the variability between cell types exceeds variability between species, confirming our results. However, in contrast to our approach, these studies[44,45] focussed on identifying commonalities in a subset of genes using conventional unsupervised clustering and statistical hypothesis testing. This is different from the approach taken here, which uses transfer learning to obtain a systems-level perspective of cell-type similarities between species. Extending our analysis beyond bone marrow to data from these recent studies is of significant interest but goes beyond the scope of this study.

Given the vast amount of single-cell data available today, integration of data from different sources is an important area of active research. A number of studies have attempted to address this issue, enabling the comparison of cells from different experiments, tissues, and species[46–48]. These methods aim to combine data via an embedding in latent space, followed by conventional analysis of cell-cell differences. Here, we follow a simplified approach, and compare similarity of cell types based on the information required to learn their representation via transfer learning, opening potential new avenues for machine learning in cell biology.

## Methods

**Mouse tissue origin**. Bone marrow from female 8-week-old C57BL/6 mice was used in this study. All experimental work including mice was approved by the Kyushu University animal experiment committee.

**Human tissue origin**. Excess marrow was collected from patients undergoing routine hip replacement surgery, with informed consent, and use of human tissue was approved by the regional ethics committee (reference 18/NW/0231).

**Bone marrow cell isolation**. Mouse bone marrow mononuclear cells (BM-MNCs) were prepared as described previously[37]. Bone marrow was flushed from tibiae and femurs and digested with 1 mg/ml collagenase IV (Thermo Fisher, 17104019) and 2 mg/ml dispase (Gibco, 17105041) in Hank's balanced salt solution (HBSS; Gibco, 14025092) for 30 min at 37 °C. Dissociated cells were treated with ammonium chloride solution to remove erythrocytes (155 mM $NH_4Cl$, 12 mM $NaHCO_3$ and 0.1 mM EDTA) for 5 min at room temperature, following 3× washes in HBSS.

Human BM-MNCs were prepared as described previously[49], with the additional removal of erythrocytes following density centrifugation through lysis in ammonium chloride solution (155 mM $NH_4Cl$, 12 mM $NaHCO_3$ and 0.1 mM EDTA) for 5 min at room temperature, following 3× washes in plain α-MEM.

**Magnetic cell sorting**. Cells were immuno-labeled with magnetic microbeads for cell separation according to the manufacturer's instructions. Up to $1 \times 10^8$ BM-MNCs were used for each separation. Human cells expressing CD45 (Miltenyi Biotec, 130-045-801) or CD235a (Miltenyi Biotec, 130-050-501) and mouse cells expressing CD45 (Miltenyi Biotec, 130-052-301) or TER119 (Miltenyi Biotec, 130-049-901) were depleted using LS columns (Miltenyi Biotec, 130-042-401) according to the manufacturer's instructions.

**Collagenase release of bone lining cells from human bone marrow**. Trabecular bone fragments obtained after the first step of cell isolation were incubated in 20 U/ml Collagenase IV (Thermo Fisher, 17104019) for 3 h at 37 °C under continuous rotation. Bone fragments were washed with plain α-MEM (Thermo Fisher, 12000-014) and cells released from extracellular matrix were filtered using a 40 μm cell strainer.

**Single-cell RNA sequencing**. Single-cell sequencing was performed as described in detail elsewhere[8] and alterations of the original protocol are reported below. Hydrophobic surface treatment of polydimethylsiloxane (PDMS) microfluidic devices was performed by incubating channels with 1% Trichloro(1*H*,1*H*,2*H*,2*H*-perfluoro-octyl)silane (Sigma-Aldrich, 448931) in Fluorinert FC-40 (Sigma-Aldrich, F9755) for 5–10 min at RT. Syringe pumps to drive both aqueous and non-aqueous phases were made in-house according to published, open source protocols[50]. Protocols for NGS library preparation described in Macosko et al. (2015) were closely followed and pre-amplification was conducted using $4 + 12$ PCR cycles (95 °C 3 min—4 cycles of: 98 °C 20 s; 65 °C 45 s; 72 °C 3 min—12 cycles of: 98 °C 20 s; 67 °C 20 s; 72 °C 3 min—72 °C 5 min; 4 °C hold). Processed libraries were sequenced using a NextSeq 500 system (Illumina) and NextSeq 500/550 High Output Kit v2 (Illumina, TG-160-2005).

**Sequence alignment**. Sequence alignment was performed as detailed in Macosko et al. (2015) using the mm10 (GSE63472) and hg19 (GSM1629193) reference genomes and STAR (version 2.5.2b) for sequence alignment. Raw reads were demuliplexed and condensed into the digital gene expression matrix (DGE) using DropSeq tools (v1.0; Macosko et al., 2015), using a modified alignment score to reduce the number of reads discarded due to multiple alignment (see "Methods: Mapping genes with multiple alignments").

**Mapping genes with multiple alignments**. The Drop-Seq pipeline implemented here utilizes the alignment output of STAR[51]. STAR provides several possible

alignment outputs for a single read (if they exist) and indicates the number of alignments for the read in the MAPQ column of the SAM/BAM alignment file. The possible Star MAPQ values are: 255—one mapping location (alignment), 3—two mapping locations, 1—three or four mapping locations, 0—five or more mapping locations. The standard Drop-Seq pipeline (as in Macosko et al., 2015) only uses unique alignments (i.e. alignments with MAPQ = 255).

The conservative viewpoint of only using unique alignments discards a large number of reads in general, but also causes gene-specific issues when the read aligns to its actual source but also to other locations. Complete rejection of these alignments is a source of gene-specific bias.

The challenge is to determine reasonable criteria for inclusion of the "correct" alignments, which we consider to be the actual source of the read. We consider that that there are conditions in which it is reasonable to retain reads that have other mappings, depending on their alignment characteristics.

Alignment characteristics (for STAR aligner):

- Reads have an AS (alignment) score from the aligner, which is the number of matching basepairs.
- The alignment that maps with the highest AS score is designated the "primary" read.
- All other alignments for the same read are "secondary" reads.
- If AS scores are the same, then the "primary" read is designated pseudo-randomly.

Every alignment from the Star aligner is tagged in the Drop-Seq pipeline by XC and XM tags (cell and molecular barcodes), an XF tag (CODING, UTR, INTRONIC, INTERGENIC), and GE tag (name of gene) if it aligns to a single gene exon in the correct orientation.

We compare the alignments for each read and flag allowed alignments according to the criteria summarized below, and set out in detail in the logic tables beneath:

- Alignments are allowed if they are exonic, have the maximum/co-maximum alignment score from the set of alignments, AND, the other alignments (with lower/equal alignment scores) are intronic/intergenic/exonic for the same gene.
- This criterion is based on the assumption that we are sequencing from mRNA and expect to align to exonic regions. However, this may not be the case, and if an intronic/intergenic alignment has a higher alignment score then we do not include an exonic alignment from the same read.
- We disallow any sets of alignments that contain mappings to different genes (as given by the GE tag).
- We do consider sets of alignments which containing more than one alignment to the same gene (if the maximum/co-maximum AS score alignment is exonic and there are no alignments to different genes). This means that a read with mappings to multiple regions on the same gene will be included in the digital gene expression output. This criterion particularly affects genes with repeat units.
- It should be noted that the same-gene criterion is suitable for digital gene expression analysis, where we are concerned with overall mRNA counts from a gene. However, the altered bam files from this pipeline should not be used in analyses for which the specific mRNA variants are import (and which use multimapping alignments). This is because the flag alterations may introduce variant specific bias.

When the multimapping criteria are met then the set of alignments is altered so that the allowed alignment is flagged as "primary" and the other alignments are flagged as "secondary". If the criteria are not met, then all alignments are flagged as "secondary".

Allowed multi-mapped reads will be included in the digital gene expression output as the DigitalExpression DropSeq program only uses "primary" alignments.

There is also a specifiable MAPQ (READMQ) threshold for inclusion in the digital gene expression output. This is set to 1 in order to allow "primary" dual, triple, and quadruple alignments. The standard dropseq pipeline uses READMQ = 10, which only allows unique mappers (which are all "primary").

**Inclusion criteria logic table for dual-, triple-, and quadruple- mapped reads.** XF tag determines whether an alignment is coding or non-coding:

- Coding if CODING/UTR
- Non-coding if INTRONIC/INTERGENIC.

If any of the XF read tags for a set of alignments is CODING/UTR but without a GE tag, then that set of mappers (dual, triple, quadruple) is not considered (all alignments set to "secondary").

Dual mapped reads: see Supplementary Fig. 6.
Triple mapped reads: see Supplementary Fig. 7.
Quadruple mapped reads: see Supplementary Fig. 8.

**Data pre-processing.** Data were analyzed using the software R (version 3.5.0) and the Seurat package (version 2.3.1). A gene mapping between mouse and humans was created using the orthologue annotation provided by Ensembl[52]. Unscaled data

were discretized (threshold > 0) and the union of genes from both species previously identified as variable (genes with mean between 0.0125 and 4; and log of dispersion >0.5) in Seurat were selected for machine learning if they were unambiguous orthologues.

**Cluster analysis.** Clustering was performed in Seurat using the Louvain algorithm[9] with resolution parameter set to 1.1.

**Cluster annotation.** To assign meaningful labels to the clusters proposed by the Louvain algorithm, differentially expressed genes were identified using the likelihood-ratio test[53] in Seurat (with settings: prevalence > 25%; fold-change > 2; $p$ value < 0.001). Obtained cluster markers were screened for previously described biomarkers for given bone marrow cell populations.

**Cell lineage tree.** A qualitative description of the cell lineage tree was obtained from the literature[13].

**Machine learning.** Machine learning models were trained using the keras for R package (v2.2.4; https://keras.rstudio.com/) and the TensorFlow backend (v1.8.0; https://www.tensorflow.org/) on a GeForce GTX 1050 GPU (NVIDIA, Santa Clara, CA, USA). To ensure robustness and protect against overfitting fivefold cross-validation was used throughout. Data were split by classes into five equal parts and five models were trained using an 80/20% training/validation split.

Because we were interested in the shared logic between the species, rather than gene expression kinetics, expression levels were binarized (1 if the gene was expressed at any level and zero otherwise) prior to learning. ANNs consisted of an input layer with 4374 units (i.e. the number of input genes), a 16-unit hidden layer with ReLU activation and a 14-unit softmax output layer with 50% dropout to the hidden layer and L1 regularization ($l = 0.001$). MLR classifiers had the same architecture, excluding the 16-unit hidden layer.

Training of all models was performed for 21 epochs with a step size of 42, and a sample generator to re-sample five training examples per class per step. Loss was calculated using cross-entropy and gradient descent optimization was conducted using RMSprop with default parameters. For training in the target domain, the step size was set to be proportional to the number of training examples up to a maximum of 30 steps per epoch. Each model from the source domain was re-trained on 1, 2, …, 10, 15, 20, 25, 30 examples per class (excluding unrepresented classes in the target domain) using fivefold cross-validation.

**Evaluation of classification performance.** To account for the extreme class imbalance, BA[54] was adapted from the binary setting described in Brodersen et al. (2010) to the multiclass setting. BA was calculated as the arithmetic mean of sensitivity (true positive rate) and specificity (true negative rate) for a given class against all other classes. Overall performance across all classes was calculated as the average BA. Further, the $F1$ score was calculated from the harmonic mean of sensitivity and precision (predicted positive rate). Performance metrics were reported as the average from fivefold cross-validation. For analyses related to Fig. 2 and Supplementary Fig. 3, classification performance of the source model in target domain was calculated from an ensemble of all five models via plurality vote. Performance in the target domain was independently assessed using a test set containing all cell not used in training (~95% of human data; range: 17.2–98.7% of human cells per class).

**Topology of misclassification.** The relationship between classes is specified by a directed acyclic graph **G**, the cell lineage tree (see Fig. 1d for reference), which is a hierarchical representation of the developmental history of cell types that was derived elsewhere (see for instance ref. [13]). In this graph, cell types are nodes and edges are direct developmental trajectories. We consider cells to be correctly classified, if the distance between the true label and the predicted label along the edges in **G** is $d = 0$. Misclassification events are categorized as proximal if the distance between true and predicted labels $d = 1$; and distal if the distance between labels $d > 1$. The proximal (and distal) misclassification rates are calculated as the number of proximal (or distal) misclassified cells divided by the number of known positive examples (from unsupervised clustering). These numbers are therefore not directly comparable to the BA, described above, which also accounts for class imbalances.

**Sensitivity analysis (MLR).** Since the MLR model consists of only an input and an output layer, and because the input features are discretized to the same range of expression values (0 or 1), for any given cell type the vector of weights of the input layer to an output class can be interpreted directly as a feature's importance to that class.

**Sensitivity analysis (ANN).** Let $X_{ij}$ be the $N \times M$ matrix of discretized gene expression values in the training dataset, where $i = 1,2,3,…,N$ indexes cells and $j = 1, 2, 3,…, M$ indexes genes, and each entry of $X_{ij} \in \{0,1\}$. Let $Y_{ik}$ be the $N \times K$ matrix of posterior probabilities for class assignments of the training samples from the ANN, where $k = 1, 2, 3,…, K$ indexes classifier classes. To enable more robust

calculation we discretized these posterior probabilities into three equal bins using the intervals [0, 1/3], (1/3, 2/3], and (2/3, 1], so that each entry $Y_{ik} \in \{1,2,3\}$.

Now fix $j = g$ and $k = c$ (i.e. consider the vector $X_{ig}$ of expression patterns of the $g$th gene in the training data and the vector $Y_{ic}$ of posterior probabilities of assignment of training data to the $c$th class by the ANN). The mutual information[55] (MI) $I_{gc}$ between these two vectors may be used to assess the extent to which knowledge of expression of the $g$th gene informs assignment to the $c$th class. In particular, a high value of the MI indicates that expression patterns of the $g$th gene are strongly associated with assignment to the $c$th class, and therefore that this gene is strongly associated with the $c$th class identity, while low value of MI (i.e. near to zero) indicates that the $g$th gene does not have a strong association with the $c$th class. For each gene $j = 1, 2, 3, …, M$ and each class $k = 1, 2, 3, …, K$ we calculated the $I_{jk}$ to obtain an $M \times K$ matrix of MI values. For each class we then ranked genes in descending order according to MI (i.e. fixing $k = c$ we ordered genes in descending order according to entries of the column vector $I_{jc}$) to obtain a list of genes ordered by the extent to which they contribute to class assignments.

**Gene set analysis**. To obtain a functional annotation of the ranked gene lists obtained from the sensitivity analyses above), gene set analysis was performed, using the 100 highest ranking genes for each class as an input to the functional annotation tool in DAVID[56] (v6.8; https://david.ncifcrf.gov/) and reference gene sets defined in the *biological process* gene ontology (GO).

**Similarity of human and mouse cell types**. To compare cell-type similarity between mouse and humans, data were normalized using the sctransform package in Seurat (v.3.1.3), accounting for differences in sequencing depth. Using the residuals obtained from sctransform, all principal components in the source domain (mouse bone marrow) were calculated and the first 16 principal components were selected as informative, using the inflection point of the scree plot as a cutoff for the dissipation of information in higher principal components. Data from the target domain (human) were projected onto the principal components of the source domain (mouse) using the loadings obtained from PCA in the source domain.

To assess similarity between cell types we first calculated the mediancentre[27] (the multidimensional equivalent of the median) of the set of expression profiles associated with each cell type, in order to establish a single expression profile that was characteristic of that cell type. Each mediancentre was calculated in the first 16 principal components (as above). To the $i$th cell in the $k$th cell type we associate a gene expression vector $G_{k,i} = (g_{k,i,1}, g_{k,i,2}, …, g_{g,i,16}) \in \mathbb{R}^{16}$, which records its status with respect to the 16 principal components we considered. Assuming that there are $n$ cells in cell population $k$, the mediancentre is that point $M_k = (m_{k,1}, m_{k,2}, …, m_{k,16}) \in \mathbb{R}^{16}$ such that $D_k = \sum_{i=1}^{n} d(G_{k,i}, M)$ is minimum, where $d_k(x,y) = \sum_{j=1}^{16} |x_j - y_j|$ is the $L_1$-distance.

For the mouse data, this yielded one characteristic expression pattern per cell type, as determined by the Louvain clustering. For the human data, this yielded one characteristic expression pattern per cell type $k$, where the membership in $k$ was determined either by (1) the Louvain clustering performed on human data or (2) by as determined by the MLR model (i.e. predicted cell identity using the mouse model).

To investigate relationship between mouse and human cell types, the cosine similarity between mediancentres was calculated for all possible pairs of human and mouse cell types.

**Human bone marrow cell characterization**. To enrich the progenitor and niche-cell subsets contained in BM-MNCs, magnetic cell sorting was employed to deplete cells expressing pan-leukocyte marker CD45 [*PTPRC*] or erythrocyte marker CD235a [*GYPA*] as well as to enrich skeletal stem cell marker STRO-1 [*HSPA8*][57]. To discriminate broad classes of cells among the BM-MNCs, unsupervised clustering was employed at low resolution in Seurat (resolution parameter = 1.1; see Supplementary Fig. 4i). This revealed the presence of 16 distinct cell types (including 5 erythroblast clusters and 2 myelocyte clusters that were each summarized as one cluster each, due to the apparent homogeneity analogous to mouse erythroblasts, compare Supplementary Fig. 1c). At this deliberately low resolution, individual genes possessed sufficient discriminative power for their identification (see Supplementary Fig. 4a–g). For instance, specific stages of neutrophil development can be identified based on the enzyme content of the secretory vesicles[58], such as primary azurophilic granules (AZU1), secondary specific granules (LTF), and tertiary gelatinase granules (MMP9), while mature neutrophils are identified based on the characteristic expression of CD16 (FCGR3B; Supplementary Fig. 4b). Moreover, CD14-positive monocytes and CD1C-positive dendritic cells[59] can be identified among monoblasts expressing versican (VCAN; Supplementary Fig. 4c). Additionally, the BM-MNC population contains a number of HSPCs, marked by the surface antigen CD34, KIT, and ANGPT1 (Supplementary Fig. 4d). Notably, a small but distinct subset of cells is marked by high levels of CXCL12 (Supplementary Fig. 4g), an important hematopoietic niche factor that, in mouse, is secreted by both osteoblasts at the endosteal surface[38] and by pericytes at the endothelial interface[60], and expression of Leptin receptor (LEPR), another marker of pericytes and adipocytes[36]. As another example, lymphocytes such as Pro-B- and Pre-B lymphocytes characterized by CD19 and CD20 (MS4A1), respectively, can be distinguished from more mature B lymphocytes marked by IgG heavy chain (IGHG2) and MZB1 (a co-chaperone important for immunoglobulin-folding[61]), respectively (Supplementary Fig. 4e, f).

**Statistics and reproducibility**. For statistical analysis and reproducibility, three biological replicates were used. This includes the use of three mice and three human bone marrow samples from separate donors. Statistical hypothesis tests were conducted using three replicates.

**Reporting summary**. Further information on research design is available in the Nature Research Reporting Summary linked to this article.

## Data availability
Data reported in this work are available from ArrayExpress under accession E-MTAB-8629 and E-MTAB-8630.

## Code availability
Computer code and machine learning models used in this study is available online from https://github.com/passt/miceandmen[62].

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

## Acknowledgements

This research was funded by the Medical Research Council (MC_PC_15078), the Research Management Committee at the University of Southampton, Faculty of Medicine and The Alan Turing Institute under the EPSRC grant EP/N510129/1. ROCO acknowledges support from the UK Regenerative Medicine Platform "Acellular/Smart Materials—3D Architecture" (MR/R015651/1), the Rosetrees Trust, Wessex Medical Research and the Biotechnology and Biological Sciences Research Council (BB/P017711/1). We would like to thank Alistair Bailey (University of Southampton) for his helpful discussion of Keras.

## Author contributions

Conceptualization: P.S.S., F.A., and B.D.M.; methodology: P.S.S., J.J.W., M.R.-Z., R.C.G.S., F.A., Y.K., and B.D.M.; investigation: P.S.S., T.N., H.I., Y.K., Y.S., and F.A.; data curation: P.S.S.; formal analyses: P.S.S. and X.D.; visualization: P.S.S. and B.D.M.; supervision: B.D.M., M.N., and K.F.; writing —original draft: P.S.S. and B.D.M.; writing —review and editing: all authors; funding acquisition: B.D.M., F.A., K.A., J.J.W., and P.S.S.; infrastructure, resources: F.A., K.A., R.O.C.O., and B.D.M.; project administration: P.S.S. and B.D.M.

## Funding

## Competing interests

The authors declare no competing interests.
