## [Peer Review File · Communications Biology]

Reviewers' comments:

Reviewer #1 (Remarks to the Author):

The authors first show that applying unsupervised clustering to scRNA-Seq data from unfractionated total bone marrow (TBM) samples and weakly lineage-depleted bone marrow (DBM) Cd45/Ter119 dual negative subsets from three different mice results in the segregation of 19 different cell types into distinct clusters. Indeed, the distance between these clusters, measured using adjacency relationships, corresponded to the known structure of the hierarchical structure of cell types in the hematopoietic cell lineages of the BM. I remark that while the authors used t-SNE to cluster their data, a simpler approach such as PCA will likely produce similar results and should be included in the paper as a baseline. PCA has the advantage that it is linear and deterministic, making interpretation much easier.

The authors then use a multi-layer perceptron with a single hidden layer of 14 units to to classify individual cells from their gene expression profiles. This model was able to reliably identify the 14 cell types that have a unique human orthologue, achieving average balanced classification accuracy of $96.7 \pm 0.9\%$ using five-fold cross validation. Given the distinct clusters achieved by unsupervised clustering, this strong performance is not surprising. It would be useful to compare the performance of the ANN with a simple classifier that uses e.g. distance in the space described by the first few (or many?) t-SNE dimensions or PCA components. This would provide a much simpler path to determine which genes are most strongly associated with each cell identity, since model coefficients could simply be read off. In general the lack of baseline models is concerning, particularly since such a simple NN performs so well at this task. Figure 1c suggests that a simple classifier that uses cosine similarity in the t-SNE or PCA embedding space will also perform well.

The authors then tackle the challenge of Human BM biology. They sequenced BM samples from three humans, obtaining gene expression profiles for 9,394 cells. As for the mouse case, unsupervised clustering yielded a classification of the data by cell type. The authors applied their mouse ANN to the human data, to ask how well it performed at classifying these cells - it performs well for some cell types, and less well for others. An important question that the authors must address is how a baseline simple classifier will perform at this task. It is completely unclear to me that there is any need for an ANN to solve this problem. For example, if they use the first few (or many) principal components to build a simple linear classifier for the mouse data, then they could immediately apply this classifier to the human data and measure performance. This would amount to identifying a linear combination of genes whose expression indicates specific cell types, where the same linear combination works for mouse and human.

The benefit of a linear model, or any other simple model, is that it is much more interpretable and understandable than any neural network. There is also much less danger of model overfitting, because fewer parameters are used. Simple approaches could also be applied to transfer the simple model from mouse to human. Essentially this involves defining a similarity metric on the space of gene expression profiles, where only genes with both mouse and human homologues are considered in the profiles, and learning for different cell types how much the profiles differ between human and mouse.

Major comments:

I urge the authors to consider writing a much simpler version of this paper, which compares the performance of different models at this task, and gets at the underlying question which is how similar the gene expression profiles are for different BM cell types between mouse and human. The authors have access to a great dataset, and carrying out the simplest possible analysis would really illuminate the similarities and differences between these profiles. Currently the use of the ANN appears largely unnecessary and obfuscates the underlying biology, which is not helpful.

Most importantly, the authors make a large appeal to the power of transfer learning in the discussion. There is nothing particularly surprising or novel about the way in which the authors have transferred between mouse and human data here. Transfer learning in this context simply involves (i) describing the cell-specific gene expression profile for a specific cell type in one species, and (ii) modifying (or not) this expected profile, or prior, using data from a few labelled cells from another species. The extent of the modification required depends on how similar gene expression profiles for these cell types are between the two species, and the amount of data needed will depend on how homogeneous gene expression profiles for this cell type are in the new species. Indeed, fig 3c shows that similar performance is achieved with 6-8 samples and no transfer learning, as with the mouse data and 0-5 human samples. I don't feel that success at this task warrants the extensive claims about 'transfer learning' that are plastered throughout the manuscript, or the extensive generalisation claims made in the discussion.

Minor comments

In the abstract the authors point out that there has been no rigorous quantification of the process of transferring biology from model organisms to man. They address this by showing that transfer learning can be used to map biology from mouse to man. It is a bit unclear to me how this mapping helps with the goal of rigorous quantification, and it would be helpful if the authors could explain this at some point in the manuscript.

I'm a bit surprised at the conclusion, at the bottom of page 9, that 'The mouse is a good model of human biology for these cell types' because the cell types have similar gene expression profiles. I would have thought (naively) that there might be other ways in which mouse and human biology might differ, that might not be captured by gene expression profiles. Is this the case?

Reviewer #2 (Remarks to the Author):

The authors develop a machine learning method to classify mouse BM cells and this method can be applied to identify human BM cells thus the approach is called transfer learning. Here are some questions.

- 1, The authors do not clearly say which gene expression matrices for ANN learning, such as TPM or counts.
- 2, Since they are using gene expressions, how many genes are used for training and what is the proportion of missing values in the single-cell datasets?
- 3, They state the successful transfer learning. Do they find any interesting things from the ANN classifier? Such as gene clusters that can be the markers for different cell types.
- 4, The potential application of the method is not clear.

We are very grateful to the reviewers for their consideration of our paper and suggestions for improvements. We have revised the paper in accordance with all their comments and we believe that the paper is substantially improved as a result. Below we provide a point-by-point response to their queries. Our responses are in blue.

Summary of major changes

The reviewers raised an over-arching query of the extent to which our results could equally be achieved by a simpler, more-interpretable linear method. We have now revised our analysis to compare it to a generalized linear model (multiclass logistic regression, MLR). Indeed, we find that we can achieve comparable accuracy with this model, and so we have substantially revised the paper to anchor the analysis on this simplified model. For context, we have also included results from our original artificial neural network in comparison with this MLR. This comparative analysis highlights the importance of considering both model performance and interpretability when developing machine learning methods for biology. We believe that this is a very strong new message that resulted directly from the prompting of the reviewers. For this we are very grateful.

Reviewer #1

The authors first show that applying unsupervised clustering to scRNA-Seq data from unfractionated total bone marrow (TBM) samples and weakly lineage-depleted bone marrow (DBM) Cd45/Ter119 dual negative subsets from three different mice results in the segregation of 19 different cell types into distinct clusters. Indeed, the distance between these clusters, measured using adjacency relationships, corresponded to the known structure of the hierarchical structure of cell types in the hematopoietic cell lineages of the BM. I remark that while the authors used t-SNE to cluster their data, a simpler approach such as PCA will likely produce similar results and should be included in the paper as a baseline. PCA has the advantage that it is linear and deterministic, making interpretation much easier.

We are very grateful to the reviewer for his careful overall assessment of our work and their comments that have helped significantly improve our manuscript. The over-arching comment that we should consider simpler, more interpretable methods is very important and well-taken. We have revised the manuscript throughout to provide this required analysis. Indeed, in accordance with this reviewer's intuition we are able to substantially refine our analysis and produce a model that is more directly interpretable.

To begin, we would like to emphasize that we only used dimensionality reduction (e.g. tSNE) to visualise the data. All clustering was performed independently of dimensionality reduction. This has now been made more apparent in the manuscript.

The suggestion to investigate principal component analysis (PCA) as an alternative linear dimensionality reduction method is very good. We have now performed this analysis (along with an additional comparison of UMAP, another dimensionality reduction method) and included it in the revised manuscript. These results are discussed on page 4 of the revised manuscript given in full in new **Figure S3**. In this instance, the clustering structure we observed was not well preserved by PCA, although it was by nonlinear methods. Although these results suggest that genomic features combine in a nonlinear way to define cell identities, the inclusion of the PCA provides extra context and so, we think, improves the manuscript.

The authors then use a multi-layer perceptron with a single hidden layer of 14 units to classify individual cells from their gene expression profiles. This model was able to reliably identify the 14 cell types that have a unique human orthologue, achieving average balanced

classification accuracy of $96.7 \pm 0.9\%$ using five-fold cross validation. Given the distinct clusters achieved by unsupervised clustering, this strong performance is not surprising. It would be useful to compare the performance of the ANN with a simple classifier that uses e.g. distance in the space described by the first few (or many?) t-SNE dimensions or PCA components.

This is an extremely important point, and one that was clearly overlooked in the first draft of the manuscript. To rectify this, we wanted to implement a method that was (1) easily interpretable and (2) able to capture the clustering structure that was not immediately apparent in PCA (i.e. to take into account inherent nonlinearities in the data). Since tSNE dimensions are not easily interpretable, projecting onto 2 dimensions (for example) using tSNE and then building a simplified classifier in this 2-dimensional space did not provide an easily interpretable model. As an alternative, we chose to implement a multiclass logistic regression (MLR) model. Multiclass (alternatively multinomial) logistic regression is a multiclass generalization of logistic regression. This method was chosen since it is a simple generalized linear method (i.e. it makes predictions based on linear combinations of inputs via a nonlinear output function) that has been shown to be as powerful as more complex machine learning methods in other biomedical contexts while maintaining superior interpretability. Importantly, while gaining the benefits model interpretability due to its simple linear structure (gene importances can simply be read-off, as noted by the reviewer in a later comment) it is also able to learn complex patterns via a nonlinear activation function.

In accordance with the reviewer's intuition this model performed as well as our original ANN. This is a very significant point that was missing from the previous paper. To make this apparent we have revised the manuscript throughout to focus on the results on the MLR, yet we have also chosen to keep the original ANN results in the manuscript (see **Figure S4 and S5** and text throughout the manuscript) in order to provide a comparative analysis. This comparative analysis highlights the importance of considering both model performance and interpretability when developing machine learning methods for biology. We believe that this is a very strong new message for the paper that resulted directly from the prompting of the reviewer. For this we are very grateful.

We also emphasise that the ANN has a hidden layer of 16 units. In Figure 1 of the original submission, the ANN architecture was described incorrectly (as having 14 hidden units), we acknowledge an error in the manuscript text, which we have corrected in the revised manuscript.

This would provide a much simpler path to determine which genes are most strongly associated with each cell identity, since model coefficients could simply be read off. In general the lack of baseline models is concerning, particularly since such a simple NN performs so well at this task. Figure 1c suggests that a simple classifier that uses cosine similarity in the t-SNE or PCA embedding space will also perform well.

Now that we have included the MLR model we can do exactly as the reviewer suggests, and feature importances can be simply read off from model coefficients. This analysis has now been included in the paper (see **Supplementary Table 1**). Feature importances from the MLR model have also been compared to those we previously inferred from our sensitivity analysis of the ANN (now included in **Supplementary Table 3**). A very good concordance was observed. This analysis is included in new **Figure S4d**. Moreover, Gene Ontology term analysis of top-ranking features from both models reveals similar enrichment of biological processes (compare **Supplementary Tables 2 and 4**). Collectively these results suggest that both models are identifying common biological mechanisms, and argue for use of the MLR over the ANN.

The authors then tackle the challenge of Human BM biology. They sequenced BM samples

from three humans, obtaining gene expression profiles for 9,394 cells. As for the mouse case, unsupervised clustering yielded a classification of the data by cell type. The authors applied their mouse ANN to the human data, to ask how well it performed at classifying these cells - it performs well for some cell types, and less well for others. An important question that the authors must address is how a baseline simple classifier will perform at this task. It is completely unclear to me that there is any need for an ANN to solve this problem. For example, if they use the first few (or many) principal components to build a simple linear classifier for the mouse data, then they could immediately apply this classifier to the human data and measure performance. This would amount to identifying a linear combination of genes whose expression indicates specific cell types, where the same linear combination works for mouse and human. The benefit of a linear model, or any other simple model, is that it is much more interpretable and understandable than any neural network. There is also much less danger of model overfitting, because fewer parameters are used.

The reviewer is absolutely correct, and we have now performed this analysis using the MLR described above. We have revised the paper throughout to focus on this generalized linear model.

Simple approaches could also be applied to transfer the simple model from mouse to human. Essentially this involves defining a similarity metric on the space of gene expression profiles, where only genes with both mouse and human homologues are considered in the profiles, and learning for different cell types how much the profiles differ between human and mouse.

This is an excellent idea. We have now included a comparison of the mouse and human cell populations exactly as suggested. Using a cosine similarity metric to determine relationships between gene expression profiles we observe that there is a strong association between comparable cell types in mouse and human. This analysis is included in **Figure 2e** and discussed on page 9.

Major comments:

I urge the authors to consider writing a much simpler version of this paper, which compares the performance of different models at this task, and gets at the underlying question which is how similar the gene expression profiles are for different BM cell types between mouse and human. The authors have access to a great dataset, and carrying out the simplest possible analysis would really illuminate the similarities and differences between these profiles. Currently the use of the ANN appears largely unnecessary and obfuscates the underlying biology, which is not helpful.

We whole-heartedly agree and have restructured the paper throughout to do this.

Most importantly, the authors make a large appeal to the power of transfer learning in the discussion. There is nothing particularly surprising or novel about the way in which the authors have transferred between mouse and human data here. Transfer learning in this context simply involves (i) describing the cell-specific gene expression profile for a specific cell type in one species, and (ii) modifying (or not) this expected profile, or prior, using data from a few labelled cells from another species. The extent of the modification required depends on how similar gene expression profiles for these cell types are between the two species, and the amount of data needed will depend on how homogeneous gene expression profiles for this cell type are in the new species. Indeed, fig 3c shows that similar performance is achieved with 6-8 samples and no transfer learning, as with the mouse data and 0-5 human samples. I don't feel that success at this task warrants the extensive claims about 'transfer learning' that are plastered throughout the manuscript, or the extensive generalisation claims made in the discussion.

The referee is absolutely right in his assessment of the transfer learning approach. The principle of transfer learning is to fine-tune the parameters of a model trained in a source domain using a small number of additional training examples from a (related) target domain. Thus, the number of training examples and extent of training needed for optimal classification performance can be substantially reduced in the target domain.

Direct comparison of naïve and transfer learning indicates that the advantage of transfer learning in this (particularly easy) learning problem is neutralized if 9 cells per class (i.e. 100 training examples) are available (**Figure S5h**). Although in the context of single-cell data, this is not an unattainable number (as demonstrated in this study), and it is expected that experiments will yield many times over the number of cells required for training. We do argue, however, that this is not always the case. For instance, if tissues with limited access are compared (for example cells from the germline) it is extremely difficult to obtain a large number of human cells due to low cell numbers and prohibitive ethical concerns. As another example, some cell types may be exceptionally rare and only a small number of cells of a certain type can be found within a tissue. In these cases, the transfer learning approach can be particularly useful.

While we only present one example of transfer learning for comparative physiology in this paper, the success of transfer learning in other areas, for instance image classification tasks has been overwhelming and we therefore strongly believe that transfer learning will be useful across many domains of biology. Furthermore, the transfer learning process itself can provide insight into similarities and differences between the source and target domains (here mouse and human bone marrow), and thus can be a tool for better understanding shared biological features, even when data is abundant in both domains. Nevertheless, we recognize that our discussion should be tempered. We have now rebalanced our discussion of transfer learning to address the reviewers concern and expanded more on the specifics of our approach and the limitations (outlined here and below).

Minor comments

In the abstract the authors point out that there has been no rigorous quantification of the process of transferring biology from model organisms to man. They address this by showing that transfer learning can be used to map biology from mouse to man. It is a bit unclear to me how this mapping helps with the goal of rigorous quantification, and it would be helpful if the authors could explain this at some point in the manuscript.

This is a valuable point that we have now tried to clarify throughout manuscript (see in particular the discussion on page 8). Essentially, we argue that the performance of the source classifier in the target domain provides an insight into the cell type mapping. We hope that this is now clearer in the revised paper.

I'm a bit surprised at the conclusion, at the bottom of page 9, that 'The mouse is a good model of human biology for these cell types' because the cell types have similar gene expression profiles. I would have thought (naively) that there might be other ways in which mouse and human biology might differ, that might not be captured by gene expression profiles. Is this the case?

This is an important point and we have now included a paragraph to the discussion to explain the limitations of our approach, including the issue of epigenetic differences raised by the reviewer.

Reviewer #2

The authors develop a machine learning method to classify mouse BM cells and this method

can be applied to identify human BM cells thus the approach is called transfer learning. Here are some questions.

We are extremely grateful to the reviewer for their consideration of our paper. We have revised the manuscript throughout to address their concerns.

1, The authors do not clearly say which gene expression matrices for ANN learning, such as TPM or counts.

We are grateful to the referee for highlighting this point as not sufficiently clear. For the machine learning part, we used discretized gene counts and we describe this in the Methods section of our manuscript (see **Methods - Data pre-processing**). In our revised manuscript, we also describe this point in the main text to improve clarity and we have made this more explicit in the Methods section.

2, Since they are using gene expressions, how many genes are used for training and what is the proportion of missing values in the single-cell datasets?

We use 4372 genes for training (see **Figure 1g**). The proportion of missing values (sparsity) in the single cell dataset is 93.25% on average in mouse (95.94% in human – this difference not significant according to a Wilcoxon rank sum test, $p=0.1$). Such sparsity is commonly observed in single-cell gene expression data (see for example, Lähnemann et al 2020 PMID: 32033589). We have now included the level of sparsity in **Figure S3e** and added to the main text of the manuscript to clarify this point (see page 1 and page 6).

3, They state the successful transfer learning. Do they find any interesting things from the ANN classifier? Such as gene clusters that can be the markers for different cell types.

In the revised manuscript we have compared the performance of the ANN from the original paper with a simpler multiclass logistic regression model. In the revised paper we have identified the genes that correlate with different cell identities for both models (see page 6 of the revised manuscript). These genes can be used as markers for different cell types. We observed a strong overlap in the markers associated with the two models, indicating that they identify the same driving biological processes. Moreover, the gene expression patterns of these important genes are broadly preserved between species (e.g. see **Fig. S4i, j**). A full list of the genes associated with each cell type, both in mouse and human, are now included in **Supplementary Tables S1-4**.

4, The potential application of the method is not clear.

In our manuscript, we demonstrate that machine learning methods can be combined with experiment to better understand biology and how it maps between the species. Thus, our results shed light on species-specific and shared biological processes. Practically, analysis of features (i.e. genes) that drive different cellular phenotypes, and how these features vary between species, can be used to define cellular populations more precisely. The MLR model that we consider in the paper is easily interpretable, and so more easily applicable than the ANN described in the original paper. We have now rewritten the discussion to make this point clear.

Reviewers' comments:

Reviewer #1 (Remarks to the Author):

The article has significantly improved with the addition of the multinomial logistic regression model.

I am not an expert in the field of single cell RNAseq data analysis, and in particular the question of cross species comparisons of this data. However, I do recall that for the context of human and mouse, there was quite some controversy over the question of whether data from the ENCODE consortium clustered by species rather than by tissue type (see <https://www.pnas.org/content/111/48/17224/>, <https://f1000research.com/articles/4-121/v1> and <https://genomebiology.biomedcentral.com/articles/10.1186/s13059-015-0853-4>, more recently <https://www.nature.com/articles/s41586-018-0590-4>). I am a bit surprised that this and related prior work does not appear to be cited in this paper. Prior findings include that this data clusters by tissue type rather than by species when human and mouse data are compared, and by cell type when data from multiple mice is combined. The novel contribution of this paper beyond the transfer learning result is not clear to this reviewer.

Furthermore, it is not at all clear to me that the authors needed to collect experimental data to obtain this result, given the large amount of data for human and mouse available in the public domain. Surely at the very least the authors can use existing data from the public domain to extend their result to other tissue/cell types? As I remark, I am not familiar with this literature, and so it may be that previous studies have not specifically considered the erythroid, myeloid and lymphoid branches of the hematopoietic lineage tree. Please could the authors clearly describe the previous work that has been carried out in this domain in the introduction to their manuscript.

Reviewer #2 (Remarks to the Author):

The questions are addressed with satisfaction.

We would like to thank the referees for their careful assessment of our work. We are very pleased that we were able to satisfy all of the referee's queries with the initial revision of our manuscript and we hope that the additional changes implemented in this second revision will resolve the remaining questions. We provide a point-by-point response to the comments below - our comments are highlighted in blue.

Reviewer #1 (Remarks to the Author):

The article has significantly improved with the addition of the multinomial logistic regression model.

We are very grateful to the referee for his positive assessment of the revised manuscript and are pleased that our changes to the manuscript have been recognized as a significant improvement.

I am not an expert in the field of single cell RNAseq data analysis, and in particular the question of cross species comparisons of this data. However, I do recall that for the context of human and mouse, there was quite some controversy over the question of whether data from the ENCODE consortium clustered by species rather than by tissue type. I am a bit surprised that this and related prior work does not appear to be cited in this paper. Prior findings include that this data clusters by tissue type rather than by species when human and mouse data are compared, and by cell type when data from multiple mice is combined. The novel contribution of this paper beyond the transfer learning result is not clear to this reviewer.

We are thankful for this excellent suggestion and we regret that we did not reference the important prior work of the cross-species comparison in our manuscript. To significantly strengthen the manuscript and set our results more firmly in context, we have now included references to previous work into the introduction of our revised manuscript and furthermore discuss the findings from these studies at length in our discussion. We believe that the addition of this prior work in the revised manuscript, has improved clarity of the motivation and novelty of our study.

Furthermore, it is not at all clear to me that the authors needed to collect experimental data to obtain this result, given the large amount of data for human and mouse available in the public domain. Surely at the very least the authors can use existing data from the public domain to extend their result to other tissue/cell types? As I remark, I am not familiar with this literature, and so it may be that previous studies have not specifically considered the erythroid, myeloid and lymphoid branches of the hematopoietic lineage tree. Please could the authors clearly describe the previous work that has been carried out in this domain in the introduction to their manuscript.

The reviewer is right in his assessment, that an increasing amount of single-cell data is now available in the public domain. For the current study it is of paramount importance to conduct experiments that are tailored to our research question and hence we designed an experimental protocol that enables the direct comparison of mouse and human data without differences in the practical implementation. This aspect limits the reuse of public data to extend our results since, for one, there are no equivalent paired human-mouse bone marrow data available and, for another, analysis of data from another tissue and providing analyses at a similar level of detail as provided here for the bone marrow would detract from our aim of understanding bone marrow biology (which is our area of expertise). However, we do acknowledge the reviewers concerns and agree that the current emphasis on a general comparison of mouse and human biology in the title may be too strong and detract from this message. To address these concerns and make our contribution clearer we plan to retitle the paper "*Mapping bone marrow biology from mouse to man using transfer learning*".

(see <https://www.pnas.org/content/111/48/17224/>, <https://f1000research.com/articles/4121/v1> and <https://genomebiology.biomedcentral.com/articles/10.1186/s13059-015-0853-4>, more recently <https://www.nature.com/articles/s41586-018-0590-4>)

Reviewer #2 (Remarks to the Author):

The questions are addressed with satisfaction.

We are thankful to the reviewer for his time and pleased that we were able to satisfy all queries.